

# Landfalling tropical cyclones significantly reduce Bangladesh's energy security

Kieran M. R. Hunt[1,2] and Hannah C. Bloomfield[3]

[1]Department of Meteorology, University of Reading, Reading, UK
[2]National Centre for Atmospheric Science, University of Reading, Reading, UK
[3]Department of Civil and Geospatial Engineering, School of Engineering, Newcastle University, Newcastle upon Tyne, UK

**Correspondence:** Kieran M. R. Hunt (k.m.r.hunt@reading.ac.uk)

**Abstract.** Bangladesh's rapidly expanding, yet fragile, electricity grid is highly exposed to tropical cyclones. However, the operational impacts of these storms on the power system are not well quantified. Here, we combine daily metered electricity demand data for Bangladesh's nine power zones with meteorological and hazard datasets over the last decade. We find that landfalling tropical cyclones cause an average 20% reduction in national electricity supply, with coastal zones disproportionately affected, experiencing drops of up to 38%. Analysis of case studies shows that high winds, storm surge, and extreme precipitation are all key contributors to these outages. While Bangladesh imports power from neighbouring West Bengal (India) to strengthen security, we show that cyclone impacts are often correlated across both regions, limiting the reliability of this backup during major events. We highlight the need for continued investment in climate-resilient energy infrastructure in the region, as well as adaptation to such extremes, which are projected to become more severe with climate change.

## 1 Introduction

Bangladesh's position at the head of the Bay of Bengal regularly exposes it to tropical cyclones and depressions. Approximately ten tropical cyclones make landfall in Bangladesh each decade (Islam and Peterson, 2009), though this is subject to substantially decadal variability. Compounding this frequent hazard, more than half of Bangladesh has an elevation below 5 metres (see Fig. 1), and approximately one quarter of the population (about 40 million people) live within 500 km of the coast. Therefore, tropical cyclones and depressions, which are often associated with storm surges exceeding 5 m (Karim and Mimura, 2008; Chiu and Small, 2016), present a serious hazard to life and infrastructure in this heavily exposed region (Hossain, 2015). Although the basin-wide frequency of tropical cyclones has slightly declined in recent decades, both observation- and model-based studies point to a rise in the frequency of more intense cyclones (Sahoo and Bhaskaran, 2016; Balaji et al., 2018; Singh et al., 2019; Baburaj et al., 2020; Knutson et al., 2020; Roberts et al., 2020). This trend has been underscored by a wealth of high-impact landfalling tropical cyclones in the last two decades, e.g., Sidr (2007), Aila (2009), Viyaru (2013), Roanu (2016), Mora (2017), Amphan (2020), Mocha (2023), and Remal (2024).

National power system models are growing ever more sophisticated with recent increases in computing power, with hourly multi-decadal simulations now possible (Grochowicz et al., 2024). Although matching supply and demand is only part of the problem, and state-of-the art developments now incorporate network damage from tropical cyclones (Bennett et al., 2021a) and





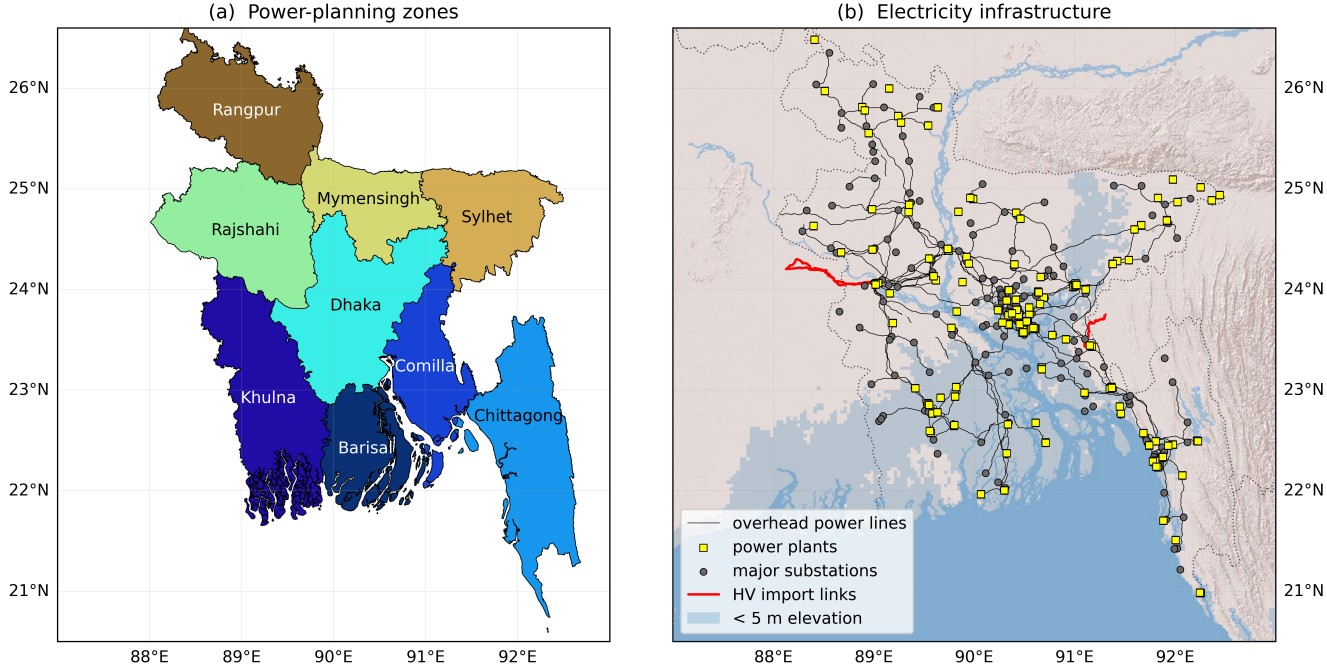

**Figure 1.** Maps showing (a) the power-planning zones of Bangladesh and (b) the layout of its core electricity infrastructure. The zones in (a) are identical to the divisions (i.e., largest administrative unit) of Bangladesh, except that Chittagong is split into two, with the northwestern half referred to as Comilla. Infrastructure data are taken from OpenStreetMap (see Sec. 2.2). © OpenStreetMap contributors 2025. Distributed under the Open Data Commons Open Database License (ODbL) v1.0.

the impacts of climate change (Wohland et al., 2025). However, understanding the resilience of systems (e.g., the amount of local damage that happens from an extreme event and the recovery time) is best understood through analysis of historical data. Tropical cyclones pose the largest environmental risk to the countries bordering the Bay of Bengal, with Bangladesh (REF) and are therefore the focus of much of the literature for the limited analysis available for our study area. As metered data is often limited, Mo et al. (2025) used satellite-based nighttime light observations across 66 countries and 396 storms and found

that the median duration of blackouts resulting from storms worldwide is four days, with longer recovery times in less dense urban areas and developing regions with less stringent energy system regulation.

In Bangladesh, these storms strike an electricity grid (Fig. 1) that, while rapidly expanding, remains structurally fragile (World Bank, 2021, 2024). Total installed power capacity stands at around 27 GW as of May 2025, a fivefold increase on ten years ago (Bangladesh Power Development Board, 2025), with over half coming from depleting resources of domestic gas.

Imported diesel, oil and coal fill most of the gap, leaving renewables (almost entirely biofuel) comprising about 15% of the mix (International Energy Agency, 2025). Bangladesh now has over 14,000 km of high-voltage transmission lines (Fig. 1(b)), increasing at a rate of about 1,000 km per year (Bangladesh Power Development Board, 2023). Tropical cyclones impact all parts of critical energy infrastructure. For example, strong winds damage or knock down transmission and distribution network



equipment (Cyclone Sidr damaged 43,000 poles and 154 towers; Bangladesh Ministry of Food and Disaster Management
(2008)), and seawater inundation from storm surges shorts or otherwise damages the many coastal substations and power
plants (Fig. 1(b)) (Shahid, 2012).

To reduce shortages, Bangladesh engages in cross-border trade with India, with capacity to import up to 1.2 GW through two
high-voltage links (with more planned) – about 1.0 GW via Baharampur–Bheramara and ∼160–192 MW via Tripura–Comilla
(an upgrade to 500 MW is underway) – both of which are shown in red in Fig. 1. Reliance on these imports, however, raises
its own hazards. Firstly, West Bengal lies in the same basin as Bangladesh and tropical cyclone footprints often cover parts of
both regions. Cyclone Amphan resulted in widespread damage and loss of life in both West Bengal and Bangladesh (World
Meteorological Organization, 2020; India Meteorological Department, 2024; Needs Assessment Working Group (NAWG) and
Humanitarian Coordination Task Team (HCTT), 2020), demonstrating that simultaneous damage on both sides of the border
could rapidly erode the diversification benefit. Secondly, because cross-border exchanges are scheduled day-ahead or week-
ahead and updated intraday subject to available transfer capability, Indian system operators may reduce the offered export in
advance when a cyclone is forecast to affect eastern India. Realtime demand-side management also remains possible if grid
security requires it (Central Electricity Regulatory Commission, 2023).

While existing literature documents cyclone mortality and economic losses, we still have limited quantitative insight into (a)
the operational, regional, and demand-side impacts of tropical cyclones on Bangladesh's electricity grid; and (b) the real-world
reliability and feasibility of importing electricity from India when both grids face correlated stress from large-scale wind, rain,
and surge impacts. In this study, we present an initial investigation into these questions by combining metered demand data
across the nine power-planning zones of Bangladesh with data on landfalling tropical cyclones and depressions over the last
decade.

## 2 Data

### 2.1 Electricity demand data

We use daily electricity demand met data – i.e., the energy actually supplied by the grid to consumers – from both Bangladesh
and West Bengal.

### 2.1.1 Bangladesh

The Bangladesh Power Development Board publish daily totals of electricity demand for each of the country's nine power
zones (Fig. 1(a)). These are archived at URLs of the form https://misc.bpdb.gov.bd/area-wise-demand?date=DD-MM-YYYY,
from which they can be readily scraped. This gives us daily data, starting in December 2015 and running through to the present.
With some days missing from the archive, and following simple quality control, we are left with about 99.7% coverage.

We have made the scraped data available in machine-friendly format at «Zenodo link». The code to download the data and
recreate all figures and analysis in this paper is available at https://github.com/kieranmrhunt/bangladesh-renewable. This is





part of a wider research group initiative to make energy data open and accessible to all, e.g., for training large-sample demand models.

### 2.1.2 West Bengal

Daily demand met data for West Bengal are taken from the Indian dataset published by Hunt and Bloomfield (2025). They extracted the data from daily reports issued by the Grid Controller of India (Grid-India), known before November 2022 as the
Power System Operation Corporation (POSOCO). The reports are in PDF format and contain the total energy supplied by the national grid to each state on each day. The final version of the dataset covers all Indian states from 2014 to present, and is available at https://zenodo.org/records/14983362.

## 2.2 OpenStreetMap

OpenStreetMap (OpenStreetMap contributors, 2017) (OSM) is a collaborative project to create a free, open, editable map of
the world. It allows users to view, edit and add data, and has a large user community that contributes data on a wide range of topics, including location names, road networks and points of interest. The completeness of the data can vary because it relies on the contributions of volunteers. In this study, we use OSM data for a brief qualitative analysis of the locations of power infrastructure in Bangladesh (Fig. 1(b)). These were extracted using Overpass Turbo (https://overpass-turbo.eu/) and extracting entities where the 'power' variable is in 'line', 'plant', 'substation', or 'tower'.

## 2.3 Weather and hazard data

### 2.3.1 ERA5

ERA5 is the fifth generation atmospheric reanalysis of global climate produced by the Copernicus Climate Change Service at the European Centre for Medium-Range Weather Forecasts (Hersbach et al., 2020). Data from ERA5 have global coverage at a horizontal resolution of ∼25 km and resolve the atmosphere on 137 levels from the ground up to ∼80 km in altitude. It
covers from January 1940 until present at hourly frequency. We use single-level data (https://doi.org/10.24381/cds.adbb2d47) to show the near-surface instantaneous 10-m vector wind fields associated with blackout case studies, as well as 'significant height of combined wind wave and swell' as a proxy for the magnitude of storm surges. There are global biases in ERA5's under-representation of the magnitude and intensity of surface wind gusts, particularly over complex terrain (Belmonte Rivas and Stoffelen, 2019), but here we use it qualitatively to contrast the relative intensities and structures of the cyclones.

### 2.3.2 GPM-IMERG

Global Precipitation Mission Integrated Multi-satellitE Retrievals for (GPM-IMERG) (Huffman et al., 2015) is a global satellite-based precipitation product at a half-hourly, $0.1°$ resolution, starting in June 2000 and continuing to the present day. Over the tropics, IMERG primarily ingests retrievals from (for 2000—2014) the now-defunct Tropical Rainfall Measuring Mission (Kummerow et al., 1998, 2000) 13.8-GHz precipitation radar and microwave imager (Kozu et al., 2001) and (for



2014 onwards) the Global Precipitation Measurement (Hou et al., 2014) Ka/Ku-band dual-frequency precipitation radar. We use IMERG precipitation data (0.1°/30 min) to quantify rainfall in our case studies because its finer resolution than ERA5 (0.25°/hourly) and demonstrated skill in reproducing the spatial and diurnal structure of precipitation over Bangladesh (Ahmed et al., 2021) make it preferable for resolving small-scale extremes; more broadly, many evaluations report IMERG outperforming reanalyses such as ERA5 in convective regimes (e.g., Xin et al., 2022).

### 2.3.3 EM-DAT

The Emergency Events Database (e.g., Delforge et al., 2025) maintained by the Centre for Research on the Epidemiology of Disasters provides a homogenised catalogue of natural hazards and their reported impacts. We queried EM-DAT for the period December 2015–May 2025 and retained all events whose affected country list contains Bangladesh. We then subset to hydrometeorological classes potentially relevant to the power system: lightning, floods, coldwaves, heatwaves, and wildfires. For tropical

cyclones and depressions, we additionally verified the reported event dates against the India Meteorological Department (IMD) Regional Specialized Meteorological Centre metadata (https://rsmcnewdelhi.imd.gov.in/report.php?internal_menu=MzM=) to ensure that the disturbance made landfall over, or had a clearly documented direct impact on, Bangladesh. The EM-DAT catalogue is used in two ways: (i) to tag and interpret sharp demand shortfalls that are not associated with a tropical cyclone or depression (Fig. 2); and (ii) to ensure that our composites of 'other' hazards do not mix qualitatively different drivers (e.g.,

major river flooding versus pre-monsoon mesoscale convective systems).

## 3 Results

### 3.1 Quantifying impacts

Energy demand in Bangladesh has increased by approximately 70% in the last ten years (Fig. 2), from an average of 7200 MW in 2016 to 12400 MW in 2024. Within this rapid growth, there is a pronounced annual cycle – with an average magnitude of

about 3300 MW – and weekly cycle – with an average magnitude of about 580 GW.

Outside of these cycles, there are short drops in demand often lasting only a day or two, which almost invariably coincide with hazardous weather events, with tropical cyclones being the biggest contributor. The most extreme case occurred during Cyclone Remal: on 28 May 2024, when the average demand met across all of Bangladesh was only 4253 MW, down 67% from 12870 MW the day before, and 71% from 14472 MW the day before. These appear to affect all zones, implying issues with

central power distribution.

The majority of these large dip events are associated with a tropical cyclone or landfalling tropical depression (Fig. 3). Of those that remain, most are associated with some other kind of natural hazard, e.g., thunderstorms – although the link between these "lesser hazards" and energy shortages are sometimes coincidental, as we shall see.



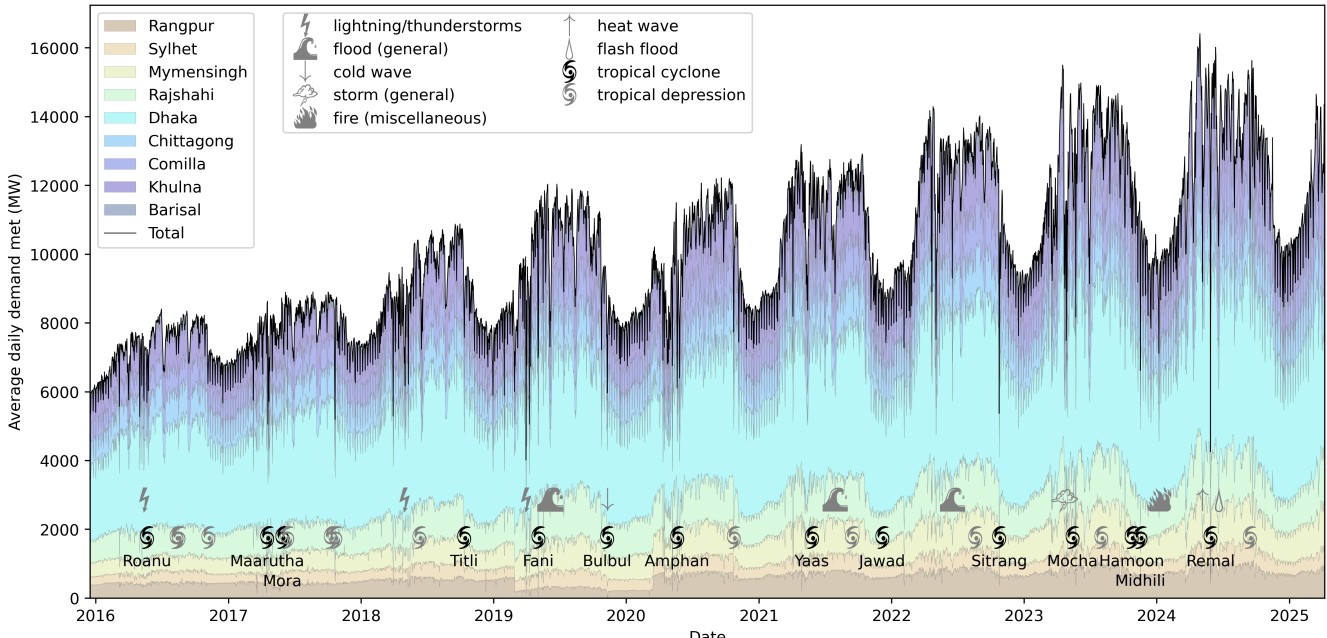

**Figure 2.** Timeline (2016–present) of average daily demand met for Bangladesh (black line) and its nine power zones (colours, see Figure 1 for locations of each region). Along the bottom, we show all tropical cyclones (with their names; black) and depressions (grey) to have made landfall in Bangladesh in this period, whether or not they coincide with power shortages. Above this, we show all other natural hazards affecting Bangladesh during this period, so long as they co-occur with an energy dip (here defined as days where national met demand falls below 80% of the 60-day running mean). Hazard data are taken from the EM-DAT archive (see Sec.2.3).

### 3.1.1 Case studies

We now extract four cases from among these large dip events. These allow us to better examine, compare, and contrast the regional structure of their causes and impacts. These cases comprise two cyclones (Remal and Sitrang), which led to the largest relative reduction in met demand; the depression with the largest reduction in met demand (and landfalling deep depression in October 2017); and a major blackout (March 2019) that did not occur alongside any significant synoptic-scale circulation, but is logged in EM-DAT as a day in which an intense thunderstorm occurred.

We start by investigating the change in regional impacts on met demand across Bangladesh's nine power zones (Fig. 4). For Cyclone Remal, we are missing data from day −2. However, on day −1, we can already see a significant impact in coastal zones, with Barisal reporting a fall of 72% in demand met and Khulna reporting a fall of 35%. On the day of landfall itself (28 May), all zones reported a large deficit. Even the least affected – Chittagong – still had a demand met of less than half of the rolling 60-day mean. After the dissipation of Remal, most zones recovered quickly, and within two days were returned to

normal power. Barisal was the exception here, having been heavily damaged by Remal, even three days later (31 May), the demand met was still 17% less than the rolling mean.





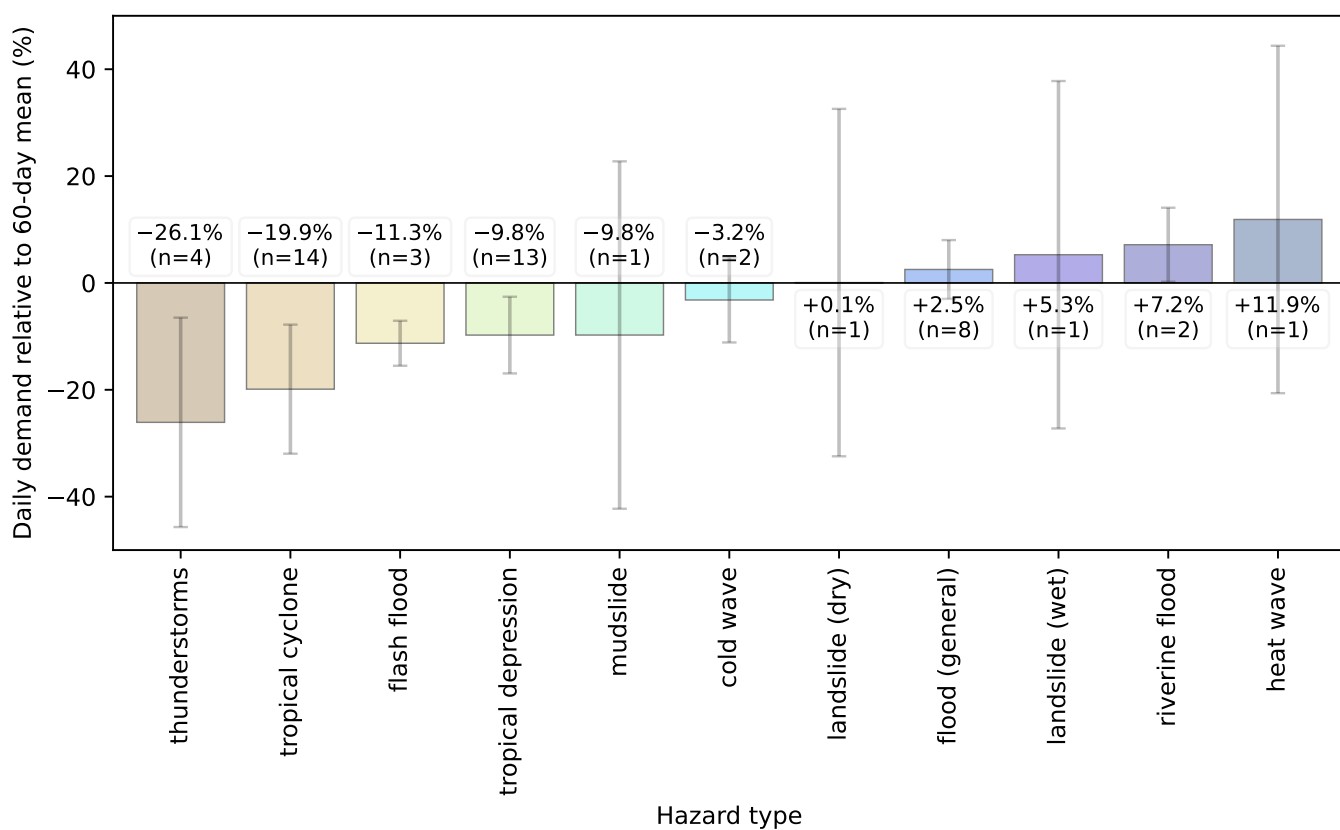

**Figure 3.** Mean day-0 change in Bangladesh national demand met relative to a centred 60-day running mean, stratified by hazard type (December 2015–May 2025). Bars show the average percentage change on the EM-DAT start date (or the cyclone landfall date for tropical cyclones/depressions), aligned to the nearest demand day within $\pm 1$ day. Error bars denote 95% confidence intervals estimated using $\pm 1.96\times$ standard error; for categories with $n=1$ the standard error uses a pooled standard deviation from categories with $n \geq 2$. Sample sizes ($n$) are shown alongside each bar. Only natural hazards are shown; non-natural classes in EM-DAT are excluded (e.g., disease or transport). However, all hazard logged for Bangladesh in EM-DAT in this period are included, whether or not they coincided with anomalous demand. Negative values indicate reductions in supply.





**Figure 4.** Relative change in met energy demand for each of the nine grid zones in Bangladesh for four case studies: (a) Cyclone Remal, (b) Cyclone Sitrang, (c) a landfalling deep depression, and (d) a major blackout not related to a synoptic-scale weather pattern. In each case, the value given is the met energy demand in each zone relative to a 60-day running mean. Day 0 is the local minimum for national met demand (for each of the cyclone cases, this is also the landfall day), and we show the previous two days and the following five. Data were not available for day −2 of Cyclone Remal or day +5 of Cyclone Sitrang. Zones are ordered (blue to brown) approximately by their distance from the coastline, starting with the nearest, consistent with earlier figures.



The profile for Sitrang is quite different. This time there is no noticeable impact on day −1. As in Remal, on the landfall day itself (24 Oct) all zones reported a large deficit (from 40% in Rajshahi to 91% in Barisal). However, the recovery was considerably slower than for Remal, such that even four days later (28 Oct), all zones reported a deficit, with all but one reporting a deficit of more than 15%. Three of the coastal zones (Barisal, Khulna, Comilla) were disproportionately affected.

The impact of the worst deep depression (Fig. 4c, landfall on 21 Oct 2017) was considerably less than either of these tropical cyclones, and the recovery was thus quicker, with all zones returning to normal just two days after landfall. As we will see in the next section, depressions often produce heavy rain much further inland than tropical cyclones. This leads to the inland zones being more affected on day 0, and the coastal zones being more affected the day before. Despite the relatively weak synoptic signature of such depressions, they are still capable of significant disruption – Mymensingh and Sylhet reported deficits of 72% and 67% respectively on the day of landfall.

The major blackout (Fig. 4d, 31 March 2019) differs from the cyclones and depression in that on day 0, there was no significant regional pattern, with neither coastal nor inland states are more affected than the other. Recovery was almost immediate (into day 1), but then a second event hit the grid (which did not conincide with any event in the EM-DAT catalogue) on day 2, again affecting a mix of coastal and inland states. This lack of specific regional impact suggests we may be able to rule out a weather-based cause, which we will now investigate.

We now present the same four cases as synoptic weather charts on the day of maximum dip (Fig. 5). Synoptic-scale storms damage electricity infrastructure through three principal hazard pathways: (i) heavy precipitation and the ensuing pluvial and fluvial flooding; (ii) storm surge and associated coastal inundation; and (iii) destructive winds. In practice these hazards co-occur, making it difficult to cleanly attribute observed power-system impacts to any single component. Bangladesh's small geographic extent relative to the typical precipitation footprint of Bay of Bengal systems further complicates attribution since most events affect the majority of the country more or less simultaneously.

Nevertheless, the three case studies in Fig. 5 do allow a degree of contrast. Cyclone Remal (Fig. 5(a)) produced comparatively modest rainfall totals over Bangladesh – the heaviest precipitation fell over north-eastern India – but generated a large storm surge, with reports of water levels exceeding three metres along parts of the coast (The Business Standard, 2024). This is consistent with ERA5 significant wave heights exceeding 6 m adjacent to Bangladesh's coastline (Fig. 5(e)), and with widespread, surge- and wind-driven damage to coastal substations and distribution infrastructure. Cyclone Sitrang (Fig. 5(b)) had weaker winds and surge, yet produced substantially greater rainfall over Bangladesh than Remal. The deep depression (Fig. 5(c)) was weaker still in terms of winds and surge, but produced even heavier rain totals across the country. This behaviour is characteristic of monsoon depressions over the Indian subcontinent, which often propagate far inland under favourable environmental conditions and are a dominant source of extreme precipitation for much of India (Hunt and Fletcher, 2019; Thomas et al., 2021). Finally, synoptic charts for the March 2019 blackout (Fig. 5(d)) confirm that this was unlikely to have been caused by weather, despite thunderstorm reports on the day.

Taken together, these events suggest a qualitative pattern. As precipitation increases and wind and surge severity decreases, the aggregate impact on the power system tends to be slightly smaller but still very large in absolute terms. All three pathways





**Figure 5.** Composite weather maps, showing instantaneous 00UTC 10-m winds (vectors) and 72-hour precipitation (accumulated from 00UTC the day before to 00UTC two days later; filled contours) for four case studies: (a) Cyclone Remal, (b) Cyclone Sitrang, (c) a landfalling deep depression, and (d) a major blackout not related to weather. (e) shows the maximum hourly signficant wave height in ERA5, relative to the event day, in a box surrounding the Bangladeshi coast (89°–92°E, 20.5°–23.0°N; marked in red in (d)) as a proxy for storm surge strength.





can produce substantial, system-wide losses on their own, and they frequently arrive in compound form. Therefore, any credible risk model for Bangladesh's national critical infrastructure must treat these hazards jointly.

## 3.2 Inter-regional complementarity

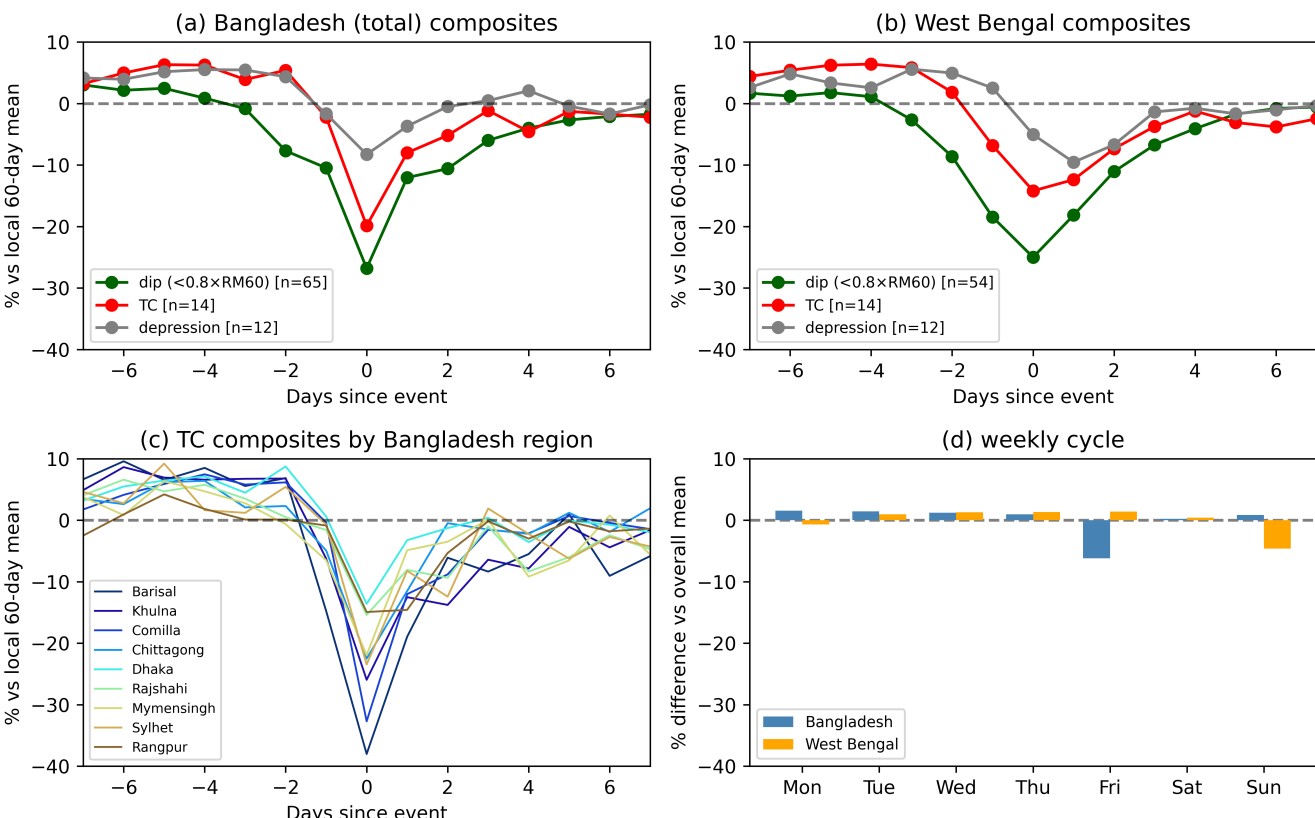

**Figure 6.** Composite impacts of extreme weather and the weekly cycle on met energy demand over Bangladesh and West Bengal. Changes in met energy demand, relative to a 60-day running mean, are averaged for landfalling TCs (red), landfalling depressions (grey), and demand 'dips' (green), and shown for (a) Bangladesh and (b) West Bengal. Dips are defined as days where met demand falls below 80% of the 60-day running mean, excluding days that fall into the TC or depression categories above. (c) the composite profile for the same TCs, separated by grid zone. (d) change in met energy demand relative to the mean, as a function of the day of week for Bangladesh (blue) and West Bengal (orange).

We now move to national-scale composite lead–lag time series of met demand, conditioned on (i) landfalling TC days, (ii)
180  landfalling depression days, and (iii) "dip" days, defined as dates on which national demand falls below $80\%$ of the two-month centred running mean (Fig. 6).





For Bangladesh (Fig. 6)a), TCs reduce met energy demand by 20% on landfall day, partially recovering to a −8% on day +1, and −5% on day +2. Depressions, similar in number over the period, have a weaker effect, reducing met energy demand by 8% on day 0, 4% on day +1, and just 1% by day +2. Of particular interest are the dip days. They are more numerous (by about a factor of 2.5) than TCs and depressions combined, and they exhibit the largest composite deficit: met demand falls by an average of −28% on day 0 and −12% on day +1.

These large composite dips are not predominantly due to weekly cycle effects as is sometimes seen for demand reductions in developed regions (e.g.(Drew et al., 2019); Fig. 6(d)). These are too weak, and Bangladesh's lowest demand day, Friday, which is when Bangladesh's energy demand is at a minimum, is associated with only about a 6% reduction relative to the weekly mean. Nor can they be attributed to public holidays. Bangladesh had 66 recognised public holidays in this period, during which demand only fell by an average of 5.5%. In fact, none of the dip days in this record coincide with national holidays in Bangladesh. Holiday composites (not shown) indicate material reductions only in the two largest urban centres: Dhaka (−11%) and Chittagong (−8%).

We see a similar pattern for West Bengal (Fig. 6b), noting that these composites are conditioned on Bangladesh landfalls. Therefore, the direct TC impact on West Bengal is weaker and temporally more drawn out (e.g., TCs reduce met demand by only about 14% on day 0. For depressions, the day of maximum West Bengal impact tends to be day +1 relative to Bangladesh landfall, consistent with the tendency for these systems to propagate westward, especially during the monsoon (Boos et al., 2015). West Bengal experiences a comparable number of dip days to Bangladesh, and in both regions their composite profiles are broader and build up over several days, in contrast to the sharper profile of TC- and depression-related dips.

The prevalence and shape of dip-day composites are sensitive to how the background is removed. If the centred running-mean window is shortened from 60 to 10 days, the slow "spin-up" largely disappears and the number of dip events falls markedly, to 35 in Bangladesh and 19 in West Bengal. This sensitivity suggests that at least a subset of dip days reflects relatively long-lived processes that depress demand over longer (e.g., weekly) timescales, such as extended flooding, heat-driven demand suppression under widespread outages, or grid-internal constraints, rather than purely short, shock-like events.

Finally, we look at the composite regional effect of TCs on the different zones of Bangladesh (Fig. 6)c). As we expect from our case study analysis earlier, coastal zones are much more severely impacted by TCs than inland zones: mean day 0 reductions reach −38% in Barisal, −33% in Comilla, and −26% in Khulna, compared with −15% in Rangpur, −15% in Rajshahi, and −13% in Dhaka. Dhaka is the least impacted zone in percentage terms, but it accounts for nearly 40% of national demand, so aggregating to the national scale inevitably downplays the severity of coastal impacts. By contrast, dip days show no systematic coastal–inland contrast (not shown), although the largest single-zone effect is again in Dhaka (−29%). There is no significant relationship between the timing of missing data and the occurrence (or lead/lag) of extreme weather.

So, although West Bengal and Bangladesh are often co-affected by the same synoptic systems, their weekly demand cycles differ (Friday versus Sunday as the principal low-demand day), so when landfall coincides with one region's weekly minimum, the joint cross-border impact can be modestly reduced.



**Figure 7.** Comparison of daily demand met in Bangladesh and West Bengal from 2016 to the present. (a) timeseries of daily demand met. (b) a scatter plot showing West Bengal demand met against Bangladesh demand met for each day in the timeseries. Outlier days are marked, with outliers defined as days where the scatter point is at least three standard deviations away from the line of best fit, or days where the sum of both Bangladesh and West Bengal demand met is in the bottom 0.1%.





### 3.2.1 Will the Bangladesh–West Bengal HV link help during shocks?

The new high-voltage exchange between Bangladesh (BD) and West Bengal (WB) is intended, in part, to provide resilience when production or transmission assets in Bangladesh are damaged. We therefore now ask: how often are Bangladesh's deepest supply shortfalls not mirrored across the border, such that imports could possibly help?

As neighbouring regions, Bangladesh and West Bengal experience very similar weather, and consequently very similar seasonal demand profiles (Fig. 7(a)). This is bad for seasonal-scale complementarity: when one system's demand is high (or low), so is the other's, limiting arbitrage value. Extreme, short-lived events are more promising. A closer inspection of the joint distribution of daily met demand (Fig. 7b) shows that large single- or few-day dips in one system are often not present in the other.

### 3.2.2 West Bengal-only deep dips are rare and cyclone-driven.

There are really only two clear episodes with very low West Bengal demand that are not matched by similarly low Bangladesh demand. The first is Cyclone Amphan (20–22 May 2020), where landfall over West Bengal led to much larger West Bengal impacts than in Bangladesh. The second is Cyclone Yaas (26–27 May 2021), where landfall over Odisha, close to West Bengal, produced a stronger West Bengal response; the system subsequently weakened before significantly affecting Bangladesh.

### 3.2.3 Bangladesh-only dips are far more common.

In contrast, there are many cases where Bangladesh met demand is markedly reduced while West Bengal is comparatively unaffected. Firstly, pre-monsoon thunderstorm activity in Bangladesh (16 June 2018, 19 June 2018, 6 May 2024, 23 May 2023), with or without an accompanying synoptic low, depressed Bangladesh met demand disproportionately. Secondly, thunderstorms over Bangladesh associated with the monsoon onset (15–20 June 2024) – during which West Bengal demand remained high, consistent with pre-monsoon heat stress there. Thirdly, there are cases with no obvious synoptic-scale driver (16–17 April 2024, 21 April 2023, 27 April 2023), where Bangladesh demand stayed high in absolute terms, suggesting temperature differences or local operational issues rather than damage.

There are also more extreme Bangladesh-only cases where weather clearly dominated, namely Cyclone Maarutha (19 April 2017), Cyclone Biparjoy (16 June 2023), and Cyclone Remal (28–29 May 2024); and several that were likely not weather-related (30 March 2018 and 31 March 2019).

### 3.2.4 Both systems low: mostly seasonal minima, not damage.

Days on which both regions reported very low met demand occur during seasonal troughs, possibly linked to public holidays: 25 December 2015, 1 January 2016, and 16 December 2019 all occurred during the climatologically coolest period of the year and are not associated with strong synoptic forcing. The notable weather-driven exception was on 10 November 2019 (Cyclone Bulbul), when both Bangladesh and West Bengal were simultaneously stressed.





## 4 Conclusions

Using almost a decade of daily, zone-resolved metered demand data, we have shown that landfalling tropical cyclones systematically and severely depress Bangladesh's electricity supply. On the day of landfall, national demand met falls by an average of ~20%, with coastal zones bearing the brunt (mean deficits up to 38%). The most extreme event in our record, Cyclone Remal (28 May 2024), reduced national demand met to 4253 MW—a 67% drop relative to the previous day and 71% relative to two days prior. Landfalling depressions are less destructive in aggregate, but still material: they reduce national demand met by about 8% on average on the day of landfall. We also identified a large class of short, sharp "dip" days unrelated to TCs or depressions that produce even larger composite deficits (28% on day 0). These dip days are too frequent and too deep to be explained by the weekly cycle (Bangladesh's Friday minimum is only ~6% below the weekly mean) or public holidays (average ~5.5% reduction), implying a mixture of other weather hazards, prolonged flood-driven constraints, operational curtailment, and grid-internal issues.

Case studies of Remal and Sitrang demonstrate that distinct hazard pathways – wind and surge for Remal; widespread heavy rain for Sitrang and the deep depression – can each, independently, generate system-wide losses. Recovery dynamics differ between events: following Remal, most zones recovered to within 5–10% of normal within two days, whereas after Sitrang, substantial deficits persisted four days later. Inland zones are consistently less affected in percentage terms, but Dhaka alone accounts for nearly 40% of national demand; aggregating to the national scale therefore masks the disproportionate coastal burden and the operational challenge of restoring service to smaller, more exposed systems.

A key practical question is whether Bangladesh's 2.5 GW of interconnection capacity with India, principally via West Bengal, offers reliable insurance during these shocks. Our analysis shows that while the links can and do provide value during many non-extreme conditions, the largest TC-induced shortfalls in Bangladesh often coincide with substantial impacts in West Bengal, because both grids are exposed to the same basin-scale hazards. Bangladesh-only deep dips are far more common than West Bengal-only dips, suggesting that, in practice, the direction of needed support will more often be from India to Bangladesh. However, precisely during the tail events when that support is most valuable (e.g., Remal-like or Bulbul-like storms), correlated stress on both systems can sharply limit the availability of imports. In other words, the diversification benefit of the existing cross-border capacity is real but bounded: it improves resilience to moderate, localised, or operational shocks, but cannot be relied upon as the primary hedge against the most damaging cyclones.

The main limitation of this study is that we work with daily totals of met demand, which conflate physical damage, deliberate load shedding, and demand suppression (e.g., due to evacuation). Finer temporal resolution (e.g., hourly dispatch) or local-level outage logs would allow a cleaner attribution of mechanisms and recovery trajectories. Our hazard characterisation is approximate (e.g., surge inference from significant wave height; coarse-resolution winds), the OpenStreetMap infrastructure layers are incomplete, and the analysis period (2015–2025) is short relative to low-frequency variability, all of which introduce uncertainty. These limitations also hinder our ability to diagnose compound and temporally compounding processes with confidence.





Our results carry clear policy relevance. Firstly, adaptation planning for Bangladesh's power system must treat wind, surge, and extreme rainfall (and flooding) as compound hazards. Hardening and elevating coastal substations, waterproofing control

equipment, or stronger tower designs in the most exposed corridors can all reduce the frequency and depth of cyclone-induced outages. Secondly, the design of cross-border imports should explicitly recognise correlated hazards, e.g., through contingency reserves. Thirdly, system operators would benefit from operational, probabilistic outage forecasts driven by weather forecasts. This would enable pre-emptive load shedding, maintenance, and the positioning of repair crews. Fourthly, distributed resources (e.g., rooftop solar,or local battery storage) offer a resilience pathway, especially for critical services, when the main electricity

grid is compromised.

Climate change will increase the frequency and severity of the climate hazards discussed previously (Knutson et al., 2020; Emanuel, 2021) with storm-surges being exacerbated in this shallow continental shelf region by rising sea levels (Rahman et al., 2019) and an increased likelihood of compound coastal flooding (Bevacqua et al., 2020). Similarly large changes in exposure, mainly driven by increasing population combined with changes hazards will further exacerbate risk (Mitchell et al.,

2022). Many infrastructure risk assessment tools are currently unable to take into account impacts of climate change as well as changes to exposure and vulnerability, and this work highlights the importance of understanding all three components within this formula for risks, as well as their associated uncertainty.

Future work could consider (i) using higher-frequency operational data and satellite night-time lights to distinguish physical damage from curtailment or demand suppression; (ii) building multi-hazard risk models that convert meteorological fields into

expected demand shortfalls; (iii) test temporal compounding by conditioning impacts on antecedent rainfall, soil moisture, or recent infrastructure faults/repairs, and by constructing storyline sequences (e.g., depressions followed by TCs, or TCs followed by heatwaves, that may result in increased air conditioning demand (Feng et al., 2022)); (iv) stress-test the grid with a national dispatch/network model (e.g., stochastically as in Boyle et al. (2022) or through optimisation modelling as in Bennett et al. (2021b)) under contrasting climate storylines and sea-level rise; and (v) disentangling the causes of non-meteorological dip

days.

In summary, Bangladesh's power system is already experiencing cyclone-driven, double-digit percentage supply shocks, and the heaviest burdens fall on its coastal zones. Cross-border links help, but cannot fully insure the system against basin-wide extremes. Building a climate-resilient electricity sector in Bangladesh will therefore demand a portfolio of measures that reduces exposure, hardens the most vulnerable nodes, decentralises critical supply, and plans explicitly for temporally

compounding, and correlated hazards on both sides of the border.

*Data availability.*

The ERA5 data used in this study can be freely downloaded from https://doi.org/10.24381/cds.adbb2d47. Similarly, the IMERG data used can be freely downloaded from https://disc.gsfc.nasa.gov/datasets/GPM_3IMERGHH_07/summary?keywords= %22IMERG%20final%22. Views of the EM-DAT database can be freely downloaded from https://public.emdat.be/. De-

mand data for Bangladesh at the zone level were scraped from daily data archived by the Bangladesh Power Development



Board (https://misc.bpdb.gov.bd/area-wise-demand). Note for reviewers: we are releasing this demand dataset as part of a larger package in the next few months. In the meantime, the complete demand data used in this study are available here: https://gws-access.jasmin.ac.uk/public/wcssp_india/kieran/bangladesh-electricity-demand.csv.

*Author contributions.* KH conceived the study and conducted the analysis, while receiving feedback from HB. KH and HB then co-wrote 315 the manuscript.

*Competing interests.* The authors declare they have no competing interests.

*Acknowledgements.* KMRH is supported by a NERC Independent Research Fellowship (MITRE; NE/W007924/1). HCB is funded by a Newcastle University Academic Track Fellowship.



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
