# Peer review of "Landfalling tropical cyclones significantly reduce Bangladesh's energy security"

_EGUsphere, 2025_

## Referee Comment (RC2)

The study integrates cyclone impacts with energy demand data — relevant and regionally novel. However, the **novelty statement** should be clearer and distinguished from prior cyclone—energy studies.

**Major comments:**

- 1. Daily demand data may mix physical outages and load-shedding. Provide **uncertainty bounds** or a brief **sensitivity test** to confirm attribution.
- 2. Include quantitative details (wind speed, surge, rainfall) for each event and test the relation between cyclone intensity and demand loss.
- 3. The India—Bangladesh power-sharing section needs quantitative support—e.g., frequency or MW impact of synchronized dips.
- 4. Adaptation options are strong but could be ranked by vulnerability or summarized in a schematic table for clarity.

**Minor comments:**

- 1. Abstract: Add numbers for average and maximum power losses.
- 2. Figures: Clarify units, improve color contrast.
- 3. Methods: Briefly mention missing-data handling and smoothing approach.
- 4. Reference: Add one or two regional energy policy sources.
- 5. Maintain consistency in "met demand" terminology.

Overall, the study is good, but improvements in quantitative precision, clarity, and uncertainty reduction are needed before acceptance.

---

## Community Comment (CC1)

Bangladesh's electricity network, though expanding quickly, remains structurally vulnerable to tropical cyclones. The extent to which these storms disrupt power system operations has not been comprehensively assessed. This study integrates daily electricity demand records from the country's nine power zones with meteorological and hazard datasets spanning the past decade. Findings indicate that cyclones making landfall reduce national electricity supply by an average of 20%, with coastal regions suffering the most—experiencing losses of up to 38%. Case analyses reveal that outages are primarily driven by strong winds, storm surges, and intense rainfall. Although Bangladesh supplements its energy security through power imports from West Bengal, India, the results show that cyclone-induced disruptions often occur simultaneously across both areas, diminishing the effectiveness of cross-border power support. The study underscores the urgency of sustained investment in climate-resilient energy systems and adaptive strategies to address the increasing severity of such extreme weather events under a changing climate. I would suggest the authors to revise the article by considering the following feedback.

Abstract: The methodology of this study is not much clear in the abstract. I would suggest to make it more explicit.

I was trying to read this section with interest, but I missed to read those recent changes and following publication on coastal region of Bangladesh.

- 1. Change in cyclone disaster vulnerability and response in coastal Bangladesh.
- 2. Factors affecting cyclone disasters death in Bangladesh

The analytical framework would benefit from greater methodological integration between the hazard and energy components. The authors present these analyses in largely parallel sections—examining meteorological intensity, storm surges, and rainfall separately from the demand-side impacts—without sufficiently articulating how these dimensions are combined quantitatively. The causal links between specific hazard metrics (e.g., wind speed, surge height, or accumulated rainfall) and observed demand reductions remain more descriptive than analytical. A more explicit coupling—through correlation analysis, regression modelling, or joint risk mapping—would strengthen the inference that particular hazard mechanisms drive grid disruptions.

Additionally, while the cross-border energy trade with India is discussed insightfully, the statistical treatment of correlated impacts could be deepened by using synchronized event analyses or network resilience modelling. The use of daily data, while pragmatic, also limits temporal precision in understanding outage onset and recovery dynamics. I am curious whether the authors could draw connections to Bhasan Char, an island located in the Bay of Bengal, Bangladesh, where solar energy is extensively utilized and has shown comparatively higher resilience during cyclone-induced disruptions.

Overall, the study makes a significant empirical contribution.

---

## Author Comment (AC1)

Manuscript Title: *Landfalling Tropical Cyclones Significantly Reduce Bangladesh's Energy Security*

Summary of the Manuscript

This manuscript examines the impact of landfalling tropical cyclones on Bangladesh's energy security, using a combination of daily metered electricity demand data for Bangladesh's nine power zones with meteorological and hazard datasets. The authors argue that landfalling tropical cyclones cause an average 20% reduction in national electricity supply, with coastal zones disproportion affected, experiencing drops of up to 38%. Finally, authors highlight the need for continued investment in climate-resilient energy infrastructure in the region, as well as adaptation to such extremes, which are projected to become more severe with climate change. The topic is timely, policy-relevant, and empirically important. The paper makes a valuable contribution to understanding the climate–energy nexus in a highly vulnerable country. It addresses a critical gap — the intersection of DRR and energy security — particularly within the West Bengal and Bangladesh context. However, the manuscript requires substantial clarification, data transparency, and necessary refinement to reach publishable standard. The argument is compelling, but methodological rigor and framing could be improved to meet international expectations.

We thank the reviewer for their positive assessment of our manuscript. We respond to their concerns in red, point-by-point below. Changes to the manuscript are marked in blue.

Major comments

- The concept of "energy security" is not clearly defined in introduction section.

We agree, we now explain it explicitly in the introduction:

"In Bangladesh, these storms strike an electricity grid (Fig. 1) that, while rapidly expanding, remains structurally fragile (World Bank, 2021, 2024), impacting its energy security. Here, energy security is used in the specific sense of security of electricity supply, i.e., the ability of the power system to deliver electricity to consumers when needed, including during short35 lived extreme events (International Energy Agency)."

We have also added text to the beginning of the methods section to contextualise the datasets we use:

"Because our focus is on operational disruption during cyclones (hours to days), we emphasise the availability and reliability components of energy security and treat daily demand met and its shortfalls as a proxy for these supply-security impacts, rather than attempting to quantify affordability or long-term adequacy (consistent with The World Bank Group, 2005)."

- Authors should clarify how cyclone impacts are mapped onto diverse dimensions.

We agree that energy security has multiple dimensions. We have clarified this by adding a short paragraph to the end of our introduction, explaining how the various components of energy security are investigated:

"Energy security is multi-dimensional and can be framed in terms of availability, affordability, accessibility and acceptability, or more generally as the vulnerability of energy systems to disruption. In this study, we present an initial investigation into these questions by combining metered demand data across the nine power-planning zones of Bangladesh with data on landfalling tropical cyclones and depressions over the last decade.

We focus on short-term electricity supply security during extreme events: (i) availability, quantified using observed shortfalls in daily demand met; (ii) reliability and resilience, quantified using the frequency, magnitude and recovery time of these shortfalls; and (iii) interdependence, assessed via the co-occurrence of deep dips in Bangladesh and West Bengal that sets the value of cross-border imports during cyclones. We do not attempt to quantify affordability or long-run adequacy."

- The introduction overlooks key comparative studies: for example, Southeast Asian energy systems and Bangladesh's coastal energy vulnerability. How this study can contribute to the existing literature?

We agree, and have added one new paragraph in the introduction explaining previous work on Southeast Asian energy systems and their response to hazardous weather; and one on the vulnerability of coastal Bangladesh:

"Recent work has quantified how extreme weather can negatively affect electricity systems across tropical coastal Asia. For Southeaast Asia, the International Energy Agency has stated that increasingly intense tropical cyclones, flooding and sea-level rise are growing threats to energy infrastructure and reliability, and urge the need for climate-aware planning (International Energy Agency, 2024). Prior estimates of cyclone-driven risks to power-grid assets across East and Southeast Asia average about 0.07% of GDP across the region and are substantially higher in some countries (0.20% in Japan; 0.17% in Laos) (Ye et al., 2024). Night-time lights have also been used to explore disruption and recovery after hazards across South and Southeast Asian countries, although reliability is limited by noise and cloud cover (Skoufias et al., 2021).

"For Bangladesh, previous studies have quantified extreme exposure to cyclonic storm surge and coastal flooding in the Ganges–Brahmaputra–Meghna delta. For example, Bernard et al. (2022) found 1-in-50-year cyclone-surge inundation heights reached ~8 m above mean sea level near the Ganges–Meghna junction, with widespread exposure of coastal islands. Recent work has further shown shows that climate hazards threaten infrastructure service delivery in coastal Bangladesh. Using a dataset of 8.2 million households, Adshead et al. (2024) estimated that, for a baseline 1-in-50-year hazard, cyclonic winds could disrupt ~94.5% of the coastal population (across essential services on average), compared with ~39.5% for coastal flooding (including storm surge) and ~22.7% for riverine flooding, with the poorest disproportionately threatened in 69% of coastal subdistricts"

- In method part, causal relationship using robust statistics is missing. I suggest to specify whether regression, correlation, or event-based impact assessment.

We agree and have clarified our approach in the revised methods section:

"Our primary methods are event-based impact assessment and compositing conditional on event types. We compute demand-met anomalies relative to a centred running-mean baseline and then analyse these anomalies around independently dated landfall events (tropical cyclones and depressions) to estimate the mean/median day-0 impact and recovery trajectory."

We have also added confidence intervals to our composite impact values:

"Using bootstrapping, the 95% confidence intervals are, for tropical cyclones [−32.0% 235 to −8.7%] and for depressions [−15.6% to −2.2%]."

We have also conducted some simple regression analysis for cyclone intensity and demand met:

"We also tested whether cyclone intensity is a useful predictor of day-0 electricity supply change. For each named tropical cyclone (Table 1), we used the IMD-reported storm class at landfall as a proxy for intensity and computed the total national day-0 demand-met deficit (again relative to the 60-day running mean). Over these 14 cyclones, deficits range from 0–69%. There is no statistically significant relationship between cyclone class and the magnitude of the day-0 deficit (Pearson $r = −0.21$, $p = 0.48$; Spearman $\rho = −0.11$, $p = 0.70$), implying that intensity class alone cannot explain the event-to-event variation in demand loss. This is consistent with the expectation that outages also depend on landfall location relative to exposed coastal infrastructure, storm size, translation speed, storm surge magnitude and location, and rainfall magnitude and location."

- I also found some missing details about data sources, for example, Cyclone dataset (JTWC, IMD, or BMD?); Energy data (BPDB, WAPDA, or open-source?); Time resolution (monthly, daily, event-based?

Our energy data sources and resolutions are already stated in Sec 2.1.1 (BPDB for Bangladesh) and Sec 2.1.2 (Grid-India for West Bengal) respectively. Both are daily.

We also already state the source of cyclone data in Section 2.3.3: "The Emergency Events Database (e.g., Delforge et al., 2025) maintained by the Centre for Research on the Epidemiology of Disasters provides a homogenised catalogue of natural hazards and their reported impacts. We queried EM-DAT for the period December 2015–May 2025 and retained all events whose affected country list contains Bangladesh. We then subset to hydrometeorological classes potentially relevant to the power system: lightning, floods, coldwaves, heatwaves, and wildfires. For tropical cyclones and depressions, we additionally verified the reported event dates against the India Meteorological Department (IMD) Regional Specialized Meteorological Centre metadata (https://rsmcnewdelhi.imd.gov.in/report.php?internal_menu=MzM=) to ensure that the disturbance made landfall over, or had a clearly documented direct impact on, Bangladesh."

No changes have been made here.

- I suggest that spatial analysis (GIS or remote sensing) could greatly strengthen the results.

We agree that further spatial analysis would strengthen our results and add further insight into hazard footprints and heterogeneous infrastructure impacts. However, the aim of this paper is

to be an empirical quantification of impacts on delivered electricity across each zone and we therefore focus on event-based frameworks. As part of this, we do already include some spatial analysis: (i) maps of Bangladesh's power zones and electricity infrastructure and (ii) meteorological composites for a set of case studies. No change has been made here.

- The conclusion is currently descriptive and too much.

We have now split this into a separate discussion and conclusions section, such that the latter can be properly devoted to summarising the paper, leaving the former to contextualise our results.

- In the discussion section, this study lacks scholarly discussion for example, how landfalling tropical cyclones significantly impacted Bangladesh's energy security

We think the reviewer is referring to our conclusions section (as our original submission did not have a discussion section). Therein, we explicitly stated: "Using almost a decade of daily, zone-resolved metered demand data, we have shown that landfalling tropical cyclones systematically and severely depress Bangladesh's demand met. On the day of landfall, national demand met falls by an average of ~20%, with coastal zones bearing the brunt (mean deficits up to 38%). The most extreme event in our record, Cyclone Remal (28 May 2024), reduced national demand met to 4253 MW—a 67% drop relative to the previous day and 71% relative to two days prior. Landfalling depressions are less destructive in aggregate, but still material: they reduce national demand met by about 8% on average on the day of landfall. We also identified a large class of short, sharp "dip" days unrelated to TCs or depressions that produce even larger composite deficits (28% on day 0). These dip days are too frequent and too deep to be explained by the weekly cycle (Bangladesh's Friday minimum is only ~6% below the weekly mean) or public holidays (average ~5.5% reduction), implying a mixture of other weather hazards, prolonged flood-driven constraints, operational curtailment, and grid-internal issues"

It is not clear what else the reviewer would like us to add.

- What is the key message for reader? Authors should connect findings to SDG 7 (Affordable and Clean Energy) and SDG 13 (Climate Action).

In our revised discussion, we now include the following subsection explicitly linking our results to these two SDGs:

"Our key result is that Bangladesh's power system is already experiencing cyclone-driven supply shocks large enough to measurably undermine short-term electricity security, and that such events disproportionately impact coastal zones while also limiting the utility of cross-border imports through correlated regional hazards. This means that continued expansion of capacity is necessary but not sufficient. Instead, ensuring that electricity is reliably delivered during extremes must be the highest priority.

This relates to both SDG 7 (Affordable and Clean Energy) and SDG 13 (Climate Action). SDG 7 is typically discussed in terms of expanding access and adequacy, but here we demonstrate the importance of the reliability of modern energy services during disasters, since large, multi-day

supply shortfalls can disrupt health care, water supply, communications, education, and livelihoods even in places that nominally have grid access.

SDG 13 focuses on adaptation and resilience. Cyclone-driven electricity disruptions of the magnitude we document here are clearly an immediate adaptation challenge for the energy sector. Resilience is a requirement for maintaining essential services during climate-related hazards and for preventing repeated disaster-driven setbacks."

Minor Comments

1. Authors clarify units for energy loss (MW vs. MWh).

We now use MW throughout.

2. I suggest to provide cyclone names and years in a new Table for reader clarity.

We agree, and have added the following table covering each of the cyclones along with their impacts to our revised manuscript:

| Cyclone | Landfall (date) | IMD category | Deaths | Damage (US$ 1 million) | Max wind speed (knots) | Min pressure (hPa) | National dip (%, max) | (date) | Zone dip (%, max) | Worst_zone (zone) | Storm surge (m, max) |
|---|---|---|---|---|---|---|---|---|---|---|---|
| Roanu | 21/05/2016 | CS | 28 | 600 | 60 | 978 | -22.3 | 21/05/2016 | -73.8 | Comilla | 6.40 |
| Maarutha | 17/04/2017 | CS | | | 50 | 985 | -33.4 | 19/04/2017 | -87.9 | Comilla | 2.52 |
| Mora | 29/05/2017 | SCS | 7 | | 80 | 963 | -3.3 | 31/05/2017 | -34.5 | Chittagong | 4.69 |
| Titli | 11/10/2018 | VSCS | | | 105 | 944 | -21.0 | 12/10/2018 | -30.3 | Sylhet | 3.08 |
| Fani | 04/05/2019 | ESCS | 39 | | 150 | 900 | -31.8 | 03/05/2019 | -78.4 | Barisal | 6.47 |
| Bulbul | 09/11/2019 | VSCS | 40 | 5.8 | 75 | 976 | -33.0 | 10/11/2019 | -97.1 | Barisal | 5.41 |
| Amphan | 20/05/2020 | SCS | 26 | 1500 | 145 | 901 | -33.5 | 20/05/2020 | -96.1 | Barisal | 9.51 |
| Yaas | 26/05/2021 | VSCS | 3 | | 75 | 970 | -3.3 | 26/05/2021 | -41.6 | Barisal | 8.72 |
| Jawad | 06/12/2021 | CS | | | 40 | 1000 | -8.0 | 06/12/2021 | -14.8 | Comilla | 1.90 |
| Sitrang | 24/10/2022 | CS | 35 | | 45 | 994 | -53.3 | 24/10/2022 | -91.4 | Barisal | 5.36 |
| Mocha | 14/05/2023 | ESCS | 3 | 1 | 145 | 908 | -11.1 | 16/05/2023 | -56.2 | Khulna | 7.10 |
| Hamoon | 24/10/2023 | VSCS | 3 | 250 | 90 | 970 | -10.2 | 24/10/2023 | -40.3 | Chittagong | 3.42 |
| Midhili | 17/11/2023 | SCS | | | 50 | 995 | -37.9 | 17/11/2023 | -89.6 | Comilla | 3.87 |
| Remal | 28/05/2024 | SCS | 16 | 90.7 | 60 | 977 | -69.4 | 28/05/2024 | -95.1 | Mymensingh | 7.06 |

**Table 1.** Summary of landfalling tropical cyclone impacts over Bangladesh (2016–2024). Landfall dates and IMD categories are from the IMD/RSMC catalogue. Deaths and damages (US$ million) are from EM-DAT where available. Maximum wind speed and minimum central pressure are taken from IBTrACS best-track data. National and zonal electricity dips are the most negative percentage anomalies in demand met relative to a centred 60-day running-mean baseline, evaluated within ±2 days of landfall. 'Worst zone' indicates the power-planning zone with the largest dip. Storm surge is taken as the maximum ERA5 significant wave height (combined wind waves and swell) within a coastal box (89–92°E, 20.5–23°N) in a −3 to +5 day window around landfall.

3. Authors should improve figure readability (especially cyclone track map).

We have added latitude and longitude markers to Figure 6, and added cyclone tracks to the precipitation/wind maps. The revised figure is shown below:

[Figure]

Figure 6. Composite weather maps, showing instantaneous 00UTC 10-m winds (vectors) and 72-hour precipitation (accumulated from 00UTC the day before to 00UTC two days later; filled contours) for four case studies: (a) Cyclone Remal, (b) Cyclone Sitrang, (c) a landfalling deep depression, and (d) a major blackout not related to weather. Tracks for the two cyclones and deep depression are given in red, with the location at each 00Z marked. (e) shows the maximum hourly significant wave height in ERA5, relative to the event day, in a box surrounding the Bangladeshi coast (89°--92°E, 20.5°--23.0°N; marked in red in (d)) as a proxy for storm surge strength.

---

## Author Comment (AC2)

The study integrates cyclone impacts with energy demand data — relevant and regionally novel. However, the novelty statement should be clearer and distinguished from prior cyclone–energy studies.

We thank the reviewer for their AI-generated review. We respond point-by-point below, in red.

**Major comments:**

1. Daily demand data may mix physical outages and load-shedding. Provide uncertainty bounds or a brief sensitivity test to confirm attribution.

We agree. The BDPB explicitly separates demand met and load shed in its tallies, so we can test this directly. Over the whole ten-year period, 18.5% of days report nonzero load shedding. Since 2022, this has increased to 45.7% of days. However, the shed load is typically very small – the median load shedding (on days where it is nonzero) is 197 MW, which is about 1% of the median demand. The 95$^{th}$ percentile of load shedding is 1301 MW, which corresponds to about 10% of the demand on those days.

Our results do not change significantly if load shedding is included. We have added a paragraph to the results section discussing this:

"Thus far, we have considered only met demand, ignoring load shedding. We now briefly discuss the impact of that choice. For cyclone landfall days, the average anomaly in met demand is –19.78%. This increases slightly to –19.48% if we add reported load shedding to obtain an upper-bound estimate of total demand (served + unserved). For depressions, it worsens slightly from –8.37% to –8.94%. Framed slightly differently, the mean absolute dip in met demand on cyclone landfall days is about 2.22 GW; mean load shedding on the same days is 116 MW. Hereafter, all results will be reported as using met demand."

2. Include quantitative details (wind speed, surge, rainfall) for each event and test the relation between cyclone intensity and demand loss.

We agree, and have added a new subsection (3.1.1 Cyclone impacts) to cover this. This comprises a table:

| Cyclone | Landfall (date) | IMD category | Deaths | Damage (US$ 1 million) | Max wind speed (knots) | Min pressure (hPa) | National dip (%, max) | National dip (date) | Zone dip (%, max) | Zone dip (zone) | Storm surge (m, max) |
|---------|-----------------|--------------|--------|------------------------|------------------------|--------------------|----------------------|---------------------|-------------------|------------------|----------------------|
| Roanu | 21/05/2016 | CS | 28 | 600 | 60 | 978 | -22.3 | 21/05/2016 | -73.8 | Comilla | 6.40 |
| Maarutha | 17/04/2017 | CS | | | 50 | 985 | -33.4 | 19/04/2017 | -87.9 | Comilla | 2.52 |
| Mora | 29/05/2017 | SCS | 7 | | 80 | 963 | -3.3 | 31/05/2017 | -34.5 | Chittagong | 4.69 |
| Titli | 11/10/2018 | VSCS | | | 105 | 944 | -21.0 | 12/10/2018 | -30.3 | Sylhet | 3.08 |
| Fani | 04/05/2019 | ESCS | 39 | | 150 | 900 | -31.8 | 03/05/2019 | -78.4 | Barisal | 6.47 |
| Bulbul | 09/11/2019 | VSCS | 40 | 5.8 | 75 | 976 | -33.0 | 10/11/2019 | -97.1 | Barisal | 5.41 |
| Amphan | 20/05/2020 | SCS | 26 | 1500 | 145 | 901 | -33.5 | 20/05/2020 | -96.1 | Barisal | 9.51 |
| Yaas | 26/05/2021 | VSCS | 3 | | 75 | 970 | -3.3 | 26/05/2021 | -41.6 | Barisal | 8.72 |
| Jawad | 06/12/2021 | CS | | | 40 | 1000 | -8.0 | 06/12/2021 | -14.8 | Comilla | 1.90 |
| Sitrang | 24/10/2022 | CS | 35 | | 45 | 994 | -53.3 | 24/10/2022 | -91.4 | Barisal | 5.36 |
| Mocha | 14/05/2023 | ESCS | 3 | 1 | 145 | 908 | -11.1 | 16/05/2023 | -56.2 | Khulna | 7.10 |
| Hamoon | 24/10/2023 | VSCS | 3 | 250 | 90 | 970 | -10.2 | 24/10/2023 | -40.3 | Chittagong | 3.42 |
| Midhili | 17/11/2023 | SCS | | | 50 | 995 | -37.9 | 17/11/2023 | -89.6 | Comilla | 3.87 |
| Remal | 28/05/2024 | SCS | 16 | 90.7 | 60 | 977 | -69.4 | 28/05/2024 | -95.1 | Mymensingh | 7.06 |

**Table 1.** Summary of landfalling tropical cyclone impacts over Bangladesh (2016–2024). Landfall dates and IMD categories are from the IMD/RSMC catalogue. Deaths and damages (US$ million) are from EM-DAT where available. Maximum wind speed and minimum central pressure are taken from IBTrACS best-track data. National and zonal electricity dips are the most negative percentage anomalies in demand met relative to a centred 60-day running-mean baseline, evaluated within ±2 days of landfall. 'Worst zone' indicates the power-planning zone with the largest dip. Storm surge is taken as the maximum ERA5 significant wave height (combined wind waves and swell) within a coastal box (89–92°E, 20.5–23°N) in a −3 to +5 day window around landfall.

and the following new text:

"Building on these composite statistics, we summarise key impacts metrics and the maximum national and zonal electricity shortfalls for each landfalling tropical cyclone in Table 1. Across the 14 landfalling cyclones in Table 1, the magnitude of the national electricity deficit ('national dip') is strongly related to the severity of the worst-affected zone ('zone dip'): within ±2 days of landfall, the maximum national dip is highly correlated with the corresponding maximum zonal dip ($r = 0.80$, $p < 0.001$), indicating that the largest national shortfalls tend to occur when at least one zone experiences near-total collapse in demand met. Correlations with human impacts are less robust, as these metrics are often missing from EM-DAT (10 of 14 cyclones have reported deaths; 6 of 14 have reported damage), but available reported deaths correlate more strongly with the worst-zone deficit ($r = −0.80$, $p = 0.006$, $n = 10$) than with the national deficit ($r = −0.56$, $p = 0.09$, $n = 10$), suggesting that mortality is more closely tied to localised, extreme disruption than to the national average. Our storm-surge proxy is moderately correlated with cyclone intensity, increasing for lower minimum central pressures ($r = −0.55$, $p = 0.04$), but the apparent correlation between reported economic damage and surge is insignificant due to small sample size ($r = 0.64$, $p = 0.17$, $n = 6$)."

3. The India–Bangladesh power-sharing section needs quantitative support—e.g., frequency or MW impact of synchronized dips.

We agree. We have conducted some additional analysis on dip synchronicity, which is in our revised results section:

"We now quantify correlated stress across the border. As before, we define 'dip' days as those when met demand falls below 80% of a centred 60-day running mean. From 2015–2025 (a total of 3393 days), Bangladesh experienced 82 dip days and West Bengal experienced 46. Only 9 days (0.27% of all days; 11% of Bangladesh dips) are synchronised (occurring on the same day) and 16 out of 82 (19.5%) Bangladesh dips coincided with a West Bengal dip within ±1 day.

During synchronised dips, the mean deficits are large in both systems (Bangladesh averaging 3.1 GW; West Bengal averaging 1.9 GW). While synchronised dips are rare, they are strongly associated with major cyclones, with 5 out of 9 occurring within ±1 day of a Bangladesh landfalling cyclone. Conditional on a Bangladesh deep dip occurring within ±1 day of cyclone landfall, the probability of a same-day West Bengal deep dip rises to ~45%, compared with ~6% for Bangladesh deep dips not occurring with a day of cyclone landfall. No synchronised dips coincided with depression landfalls."

4. Adaptation options are strong but could be ranked by vulnerability or summarized in a schematic table for clarity.

We have replaced this paragraph with a table in which we now discuss the vulnerabilities and corresponding adaptations. These are ordered by priority.

**Table 1.** Vulnerability-ranked adaptation options for Bangladesh's power system under tropical-cyclone hazards. Priority reflects the study's observed impacts (largest deficits in coastal zones; system-wide drops under compound wind–surge) and implementation feasibility.

| Priority tier | Vulnerability addressed | Example adaptations | Notes and example metrics |
|---|---|---|---|
| Tier 1 (Highest) | Coastal substations and distribution exposed to surge, seawater inundation, and high winds | Elevate and flood-proof substations, pressurise control rooms; raise equipment foundations (plinths); corrosion-resistant hardware. | Directly targets nodes consistent with largest coastal deficits. *Metrics:* Post-event recovery time; reduced demand shortfall in coastal zones. |
| Tier 2 | Wind-driven vulnerability in transmission and distribution | Targeted tower reinforcement; foundation protection (e.g., concrete cladding); regular reconductoring (i.e., replacing wires on overhead lines); redundancy in coastal regions; right-of-way vegetation management. | Focus on areas where loss creates system-wide drops. *Metrics:* Line-outage rate per storm; ensure network can tolerate loss of single critical line/substation during cyclones. |
| Tier 3 | Operational vulnerabilities, e.g., delayed response and avoidable curtailment | Probabilistic demand/outage forecasting; pre-positioning of repair teams; black-start preparedness; critical-load prioritisation protocols. | Relatively low cost, high impact. *Metrics:* Forecast reliability; time-to-restore critical loads; reduction in day-0 dip magnitude. |
| Tier 4 | Loss of critical services during widespread grid failure | Distributed solar and batteries for shelters and hospitals; microgrids in high-risk coastal communities; grid-forming inverters. | Focuses on resilience rather than supply. *Metrics:* Hours of critical-service continuity during grid outage. |
| Tier 5 (Limited during basin-wide extremes) | Cross-border imports limited by correlated BD–WB hazards | Explicit contingency reserves; diversified interconnection points; joint restoration drills; shared situational awareness. | Useful for moderate/local impacts; less reliable for basin-wide extreme weather. *Metrics:* Import availability conditional on Bangladesh dip days. |

**Minor comments:**

1. Abstract: Add numbers for average and maximum power losses.

We agree, and following a comment from another reviewer have entirely revised our abstract:

"Bangladesh's rapidly expanding electricity grid is highly vulnerable to tropical cyclones, yet operational impacts remain poorly quantified. In this paper, we investigate the impact of landfalling tropical cyclones and depressions on Bangladesh's energy security by combining daily reported demand met (across nine power-planning zones; December 2015 to May 2025) with cyclone track data and hazard proxies from reanalysis and satellite products. We use an event-centered composite approach for 14 named landfalling cyclones and 13 landfalling depressions, defining deficits in demand met as a percentage anomaly relative to a 60-day running mean. On cyclone landfall days, national demand met falls by an average of 19.8% , with the maximum recorded national deficit (69%) occurring during Cyclone Remal (28 May 2024). Coastal zones are disproportionately affected, with mean day-0 zone deficits of up to 38% and some events exceeding 90%. Depressions are associated with smaller, but still significant, deficits, averaging 8.3%. For named cyclones, the magnitude of the national deficit is strongly correlated with the worst-affected zone deficit (r = 0.80, p < 0.001), indicating that national-scale shortfalls are dominated by near-collapse in at least one zone. Cross-border analysis with West Bengal shows that the largest cyclone-related deficits are often synchronised across both regions, limiting the reliability of imported electricity during major stress events. We discuss potential mitigation and adaptation policies, such as targeted hardening of coastal network assets and decentralised backup supply for critical services as cyclone-related hazards continue to intensify under climate change."

2.  Figures: Clarify units, improve color contrast.

We now explicitly state the units in figure captions (although they were originally clearly stated in the figures themselves). It is not clear what the reviewer means by "improve color contrast" – we use standard matplotlib and/or colour-blind friendly (i.e., readable in monochrome) colour maps. No change has been made here.

3.  Methods: Briefly mention missing-data handling and smoothing approach.

We have updated our methods section (specifically the section on Bangladesh data) so that it now reads:
"This gives us daily data, starting in December 2015 and running through to the present. Some dates are missing from the archive, and some zone entries appear as zeroes, which we treat as missing data (rather than true demand). For national totals and all subsequent composite analyses, we conservatively exclude any day where at least one of the nine zones is missing (i.e., reporting zero demand), so that the national total is always based on complete zone coverage. This leaves us with approximately 99.7% coverage. To identify and analyse dips in demand met, we use a centred 60-day running mean (requiring at least 30 valid days to compute the baseline). We then express anomalies as fractional deviations from this mean and define 'dip days' as those where the anomaly is less than 80% of the running mean."

4.  Reference: Add one or two regional energy policy sources.

It is not really clear what the reviewer wants us to add, especially as we already cite several policy documents. Nevertheless, we have added the following:

"Bangladesh now has over 14,000 km of high-voltage transmission lines (Fig. 1(b)), increasing at a rate of about 1,000 km per year (Bangladesh Power Development Board, 2023). National planning is articulated in the Integrated Energy and Power Master Plan (IEPMP; Power Division, Government of Bangladesh, 2023)."

"Indian system operators may reduce the offered export in advance when a cyclone is forecast to affect eastern India. Realtime demand-side management also remains possible if grid security requires it (Central Electricity Regulatory Commission, 2023). Cross-border scheduling, access, and curtailment provisions are set out in India's Guidelines for Import/Export (Cross Border) of Electricity (Ministry of Power, Government 55 of India, 2018). At the regional level, the SAARC Framework Agreement for Energy Cooperation (Electricity) provides a multilateral policy frame for voluntary cross-border electricity trade among South Asian states (South Asian Association for Regional Cooperation (SAARC), 2014)."

5.  Maintain consistency in "met demand" terminology.

We agree. In our revised manuscript, we have replaced 20 instances of "demand met" , 7 instances of "met energy demand", and two cases of "electricity supply" with "met demand".

In the data section, we have replaced the first sentence ("We use daily electricity demand met data -- i.e., the energy actually supplied by the grid to consumers -- from both Bangladesh and West Bengal") with "We use daily demand met (served load), i.e. the electricity supplied by the grid to consumers. For Bangladesh this is reported as daily average load (MW); for West Bengal the source reports daily energy supplied (MU day$^{-1}$), which we convert to a daily-average MW equivalent where required."